# Indistinguishable mitochondrial phenotypes after exposure of healthy myoblasts to myalgic encephalomyelitis/chronic fatigue syndrome or control serum

Audrey A. Ryback[1]*, Charles B. Hillier[2], Camila M. Loureiro[3], Chris P. Ponting[1], Caroline F. Dalton[4,5]

**1** MRC Human Genetics Unit, Institute of Genetics and Cancer, University of Edinburgh, Edinburgh, United Kingdom, **2** c/o Action for ME, Bristol, United Kingdom, **3** Department of Neurosciences and Behaviour, Ribeirão Preto Medical School, University of São Paulo, São Paulo, Brazil, **4** Biomolecular Sciences Research Centre, Sheffield Hallam University, City Campus, Sheffield, United Kingdom, **5** Advanced Wellbeing Research Centre, Sheffield Hallam University, Olympic Legacy Park, Sheffield, United Kingdom

* aryback@ed.ac.uk

## Abstract

Myalgic Encephalomyelitis (ME) / Chronic Fatigue Syndrome is a disease of uncertain aetiology that affects up to 400,000 individuals in the UK. Exposure of cultured cells to the sera of people with ME has been proposed to cause phenotypic changes in these cells *in vitro* when compared to sera from healthy controls. ME serum factors causing these changes could inform the development of diagnostic tests. In this study, we performed a large-scale, pre-registered replication of an experiment from Fluge et al (2016) that reported an increase in maximal respiratory capacity in healthy myoblasts after treatment with serum from people with ME compared to serum from healthy controls. We replicated the original experiment with a larger sample size, using sera from 67 people with ME and 53 controls to treat healthy cultured myoblasts, and generated results from over 1,700 mitochondrial stress tests performed with a Seahorse Bioanalyser. We observed no significant differences between treatment with ME or healthy control sera for our primary outcome of interest, oxygen consumption rate at maximal respiratory capacity. Results from our study provide strong evidence against the hypothesis that ME blood factors differentially affect healthy myoblast mitochondrial phenotypes *in vitro*.

## Introduction

Research into Myalgic Encephalomyelitis (ME), sometimes referred to as Chronic Fatigue Syndrome, suffers from a lack of replicability that has stymied scientific progress in and consensus on this disease [1,2]. Despite the disease affecting up

**Data availability statement:** Data for the primary analyses are found in the Supporting information S2 File (OCR measurements), S3 File (ECAR measurements), and S4 File (cohort characteristics). Due to the size of the cohort and known location of the sampling, some of the information that could potentially identify participants has been redacted (BMI, age, ethnicity, comorbidities). If there are less than 5 individuals in a given category, even anonymised data should be treated as sensitive and potentially identifiable, as per guidelines outlined by the Office of National Statistics (see Review of the Dissemination of Health Statistics: Confidentiality Guidance, 2005). In order to comply with Sheffield Hallam University's ethics (ER39973246) and Article 5(1)(c) of the UK GDPR, we have redacted variables where this was the case. The ethics team at Sheffield Hallam University can be contacted at: hwbethics@shu.ac.uk.

**Funding:** This work was funded by Action for M.E. as part of a Clare Francis Research Fellowship awarded to AAR. The funder did not play a role in the study design, data collection and analysis, decision to publish, or preparation of the manuscript. Link to funder's website (which also references this specific award): https://www.actionforme.org.uk/research-campaigns/our-research-work/funding-research-and-supporting-young-researchers/.

**Competing interests:** The authors have declared that no competing interests exist.

to 404,000 individuals in the UK [3], there are no diagnostic tests, and the aetiology of the disease remains uncertain. Developing a reliable diagnostic of ME was voted priority three by the ME Priority Setting Partnership [4] and understanding how mitochondria are affected in ME was the tenth priority.

One of the most promising leads in ME research relates to phenotypic changes in both primary cells and human cell lines that have been exposed to serum from people with ME (pwME). In a study by Fluge *et al.* (2016) [5], Human Skeletal Muscle Myoblasts (HSMM) were cultured in media substituted with either healthy control sera or ME sera before undergoing a mitochondrial stress test using Agilent's Seahorse Bioanalyser platform. The assay measures the oxygen consumption rate (OCR) and extracellular acidification rate (ECAR) over sequential disruption of the oxidative phosphorylation pathway. The authors observed higher OCR under all measured conditions and higher ECAR under conditions of aerobic and anaerobic strain in myoblasts cultured in sera from pwME compared to those cultured in sera from healthy controls (HC). In particular there was a large increase in maximal respiratory capacity (MRC) in myoblasts cultured in sera from pwME. Despite the small sample size (12 pwME and 12 HC), the magnitude of the differences reported is among the largest reported in the ME literature to date: the difference in average OCR at maximal respiratory capacity had a Cohen's D = 1.32, a very large effect size [6]. Other small-scale experiments have lent credence to the hypothesis that ME sera and plasma impact cellular phenotypes. Increased mitochondrial fragmentation has been reported in human bone osteosarcoma epithelial cells grown in ME serum compared to cells grown in healthy control serum [7]. Nitric oxide production was impaired upon stimulation with G-protein coupled receptor agonists in human vascular endothelial cells treated with ME sera [8]. Fibrinaloid microclots have been reported in ME and long covid platelet-poor plasma [9], and upregulation of an autophagy factor (ATG13) was reported in the sera of pwME [10]. Recently, a study including 1,455 people with ME in the UK Biobank identified hundreds of blood traits that differ between pwME and controls [11], albeit with small effect sizes, among which could be blood factors that drive these cell phenotype changes.

These studies suggest that exposure of cell lines to serum or plasma from pwME results in phenotypic changes to the cell. However, there have been no attempts to replicate these findings, and inferences drawn from all of these studies have been limited by small sample sizes. Small sample sizes reduce the confidence of findings because they provide reduced statistical power (inability to discover a true effect), results are prone to random fluctuations (more likely to generate false positive, false negative results and overestimate effect size), and they do not reflect results from the broader population (difficulty in generalization) [12]. Confirming whether changes in cellular phenotype in healthy human myoblasts exposed to ME serum compared to control serum reflect true biological difference is a foundational step in establishing a firm evidence base to develop diagnostics or understand biological mechanisms in ME in future studies.

Here, we performed a statistically well-powered replication study of the Seahorse Mitochondrial Stress test described by Fluge *et al.* (2016). We chose to replicate this

study for four reasons: (i) Its effect size is large; (ii) Its effect is measured using a well-characterized assay (Seahorse Mito Stress Test); (iii) The signal it captures is biologically interpretable; and, (iv) Its assay is run in a 96 well format, and is therefore scalable – enabling the assay to be run on a large number of samples.

## Methods

### Samples

Participants were recruited from the Sheffield (UK) community via social media and screened for ME using a version of the DecodeME questionnaire [13] (https://osf.io/rgqs3), modified to include answer options for healthy controls, and to screen for pregnancy. All participants provided written informed consent to provide their data and blood samples for the study. Ethical approval for the project was obtained from the ethics committee of Sheffield Hallam University under the ethics number ER39973246.

Due to the female preponderance of ME [3] and to reduce heterogeneity, all study participants were female and self-reported that they were not pregnant at the time of sampling. People with ME met the Canadian Consensus Criteria (CCC) [14] and/or the Institute of Medicine (IoM) [15] diagnostic criteria and reported a clinical diagnosis of ME by a healthcare professional (Fig S1A in S1 File). In our cohort 66 pwME met both the CCC and IoM criteria, and 1 pwME met only the IoM criteria. Healthy controls did not meet the CCC or IoM criteria according to their screening survey responses, had not been diagnosed with ME by a healthcare professional, and did not report any of the 21 active comorbidities screened for by the DecodeME screening questionnaire [13]. Disease severity was based on self-reported severity scores from the DecodeME screening questionnaire, as defined in the National Institute for Health and Care Excellence guidelines for ME [16]. Characteristics, including comorbidities, of this case cohort mirror what is reported in DecodeME [17] (Fig S1B in S1 File).

Sera from pwME and HC were collected between 27/11/2023 − 23/02/2024 across two rounds of sampling over two weeks in November–December 2023 ("batch 1"), and three weeks in February 2024 ("batch 2") (Table 1).

**Table 1.  Sampling batches, demographic details and severity of pwME (ME) and healthy controls (HC).**

*Participant group*

|  | ME | HC |
|---|---|---|
| Batch 1 | 48 | 12 |
| Batch 2 | 19 | 41 |
| Age in years (median + [IQR]) | 42.0 [33.0-55.5] | 42.0 [32.0–50.0] |
| Body mass index, BMI (median + [IQR]) | 25.4 [22.7 - 29.8]* | 23.2 [21.5-25.7]** |
| *Ethnicity* |  |  |
| Asian | 1 (1.5%) | 4 (7.0%) |
| Black | 2 (3.0%) | 0 (0.0%) |
| Mixed | 1 (1.5%) | 2 (4.0%) |
| White | 63 (94%) | 47 (89%) |
| **Total** | **67** | **53** |
| **Disease severity** |  |  |
| Mild | 17 | N/A |
| Moderate | 44 | N/A |
| Severe | 5 | N/A |
| Very severe | 1 | N/A |

* missing data: 10 observations.

**missing data: 9 observations.

## Sample processing

Serum samples were collected in two red-topped VACUETTE® 6 ml CAT Serum Clot Activator (#456092) tubes, left to clot at room temperature for 45 minutes and spun for 10 minutes at 1500 g at 4°C (5 acceleration/ 5 deceleration). Serum was transferred immediately in 500uL aliquots into 1.0 ml cryotubes (#E3110-6112, Starlab) and kept on ice until transferred to the freezer. Samples were stored at −80°C until used.

## Cell culture

Human skeletal muscle myoblasts were obtained from Lonza (#CC-2580, lot number 21TL138913). Cell culture was commenced and maintained according to the manufacturer's protocols using SkGM-2 Medium (CC-3244) (bioscience.lonza.com/lonza_bs/GB/en/download/product/asset/29428). Myoblasts were kept below passage number 10 for all experiments, as reported in Fluge et al (2016).

## Seahorse mitochondrial stress tests

HSMM were seeded at 8,000 cells per well and kept in a 37°C incubator with 5% $CO_2$ overnight. The following day, cells were washed once with PBS and media changed to serum-free HSMM media supplemented with 20% serum from either a pwME or a healthy control, with media and serum refreshed on day 3. Detailed protocols can be found in the pre-registration (https://osf.io/qwp4v, 02/08/2024). On day 6, myoblasts underwent a mitochondrial stress test, performed as per the manufacturer's protocols. For the stress test 10 mM glucose, 2 µM oligomycin, 2 µM FCCP and 0.5 µM rotenone/antimycin A were sequentially added to the media of the cells and changes in oxygen levels measured using an Agilent Seahorse XF Pro bioanalyser. After the run, cells were stained with Hoechst for automated cell counting using the Cytation 1 imager interfaced with the Agilent Seahorse XF Pro.

The experiment was performed blinded and randomised: sample sets of 10 cases and 8 controls were randomised on each plate, except for the final sample set with 7 cases and 5 controls. Participant serum was applied to 5 technical replicate wells per plate. On each plate four "background" wells, without cells, were measured. In each well 3 OCR and ECAR measurements were taken under each condition: basal (amino acids), basal + glucose, proton leak, maximal respiratory capacity, and non-mitochondrial respiration, yielding a total of 15 timepoints. To account for plate-to-plate variation, 3 plate replicates were performed for each sample set. To account for possible well-to-well variation (positional effects), sample layouts were altered across these 3 plate replicates. To account for possible plate edge effects, each sample was applied to only one of the edge wells (A2-A11, H2-H11, B1-F1, B12-F12) in one of the three plate replicates. If out of the 5 wells treated with a participant's serum, 2 or fewer wells yielded useable measurements on a particular plate, additional measurements were obtained by applying the participant's serum to otherwise unused wells in sample set G. We performed an additional plate replicate for sample set C, due to wells treated with several participant sera having multiple failed measurements in that sample set. Consequently, at least 15 wells were measured per participant. To minimise technical variation, the Seahorse assay was performed using an automated liquid handler (Agilent Bravo).

## Data preprocessing

Measurements were exported from the Agilent Seahorse Analytics (https://seahorseanalytics.agilent.com) platform as.xlsx files and pre-processed using the pandas library [18] and custom scripts in Python (version 3.11.4). For each plate, the 4 measurements taken in the background wells were averaged at each of the timepoints (T1-T15) and subtracted from the measurements taken from wells with treated cells at those timepoints. For OCR analysis, the oxygen consumption rate measurements taken after rotenone/antimycin A addition (timepoints T12-T15) were subtracted from the other measurements for that well.

## Data analysis

Our analysis plan was pre-registered on August 2, 2024 on the Open Science Foundation (OSF): https://osf.io/qwp4v. Thresholds for data exclusions were decided before unblinding. We excluded wells with cell counts below 8000 and above 35000 cells as well as OCR or ECAR measurements taken at maximal respiratory capacity above Q3 + 1.5 x IQR and below Q1 – 1.5 x IQR. Wilcoxon Rank-Sum tests and Pearson correlations were performed using base R (version 4.2.2) and mixed effects models run using the lme4 package [19] and tested using lmerTest [20]. Plots were generated using ggplot2 [21] and ggbeeswarm [22] libraries. Our primary outcome was analysed with the pre-defined model1: *maximal_respiratory_capacity ~ group + scale(cell_counts) + (1 | sample_ID) + (1 | plate_id),* where "sample_ID" refers to a unique study participant. Our prediction was that the "group" coefficient would indicate higher maximal respiratory capacity in the ME group. Repeatability of the OCR at maximal respiratory capacity between technical replicates treated with the same participant's serum was calculated by running the following model: *maximal_respiratory_capacity ~ (1 | sample_ID),* and dividing the variance estimated from "sample ID" (unique study participant) by the total variance. OCR and ECAR data can be found in the supporting information S2 File and S3 File, and cohort characteristics in the S4 File. R analysis scripts can be found in the S5 File.

## Results

### Cohorts

A total of 67 pwME and 53 controls were recruited to participate in this study. A participant's blood sample was collected in either one of two rounds of sampling that took place approximately three months apart (Table 1). Cohort features were comparable across the case and control groups. All participants were female, and there was no statistically significant difference in age between cohorts (Table 1, Fig 1A). Most participants were of white ethnicity (94% ME cohort and 89% HC cohort) (Table 1). BMI was slightly increased in the ME cohort (p = 0.019) (Fig 1B). BMI was calculated from self-reported weight and height which are known to be subject to reporting bias [23], and should be interpreted with caution; reporting bias, however, is not expected to differ between the two groups. Most cases reported moderate ME (Table 1).

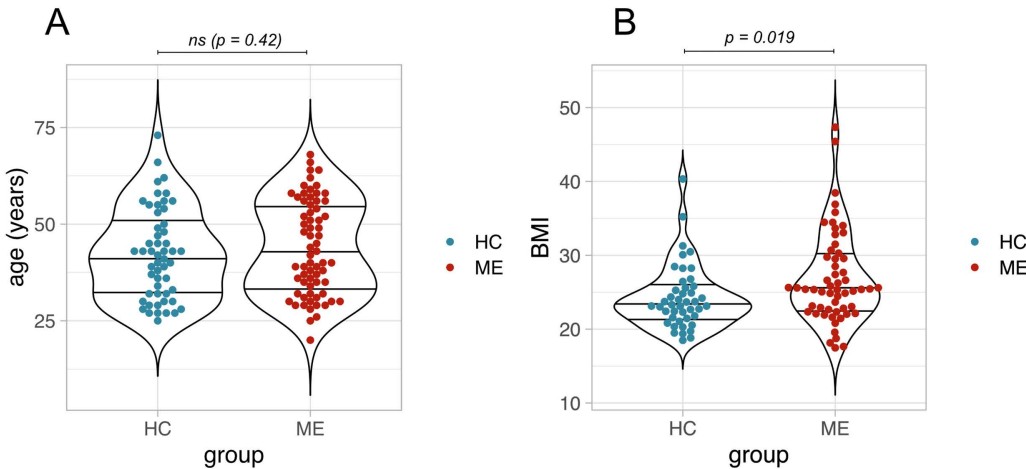

**Fig 1. Cohort characteristics. (A)** Distribution of ages in years for ME and HC cohort. **(B)** Distribution of BMI for ME and HC cohort. Differences in median age and BMI were tested using the Wilcoxon Rank-Sum Test, alpha = 0.05; ns = not significant.

## Experimental design

We hypothesised that the OCR at maximal respiratory capacity from myoblasts treated with ME sera would be higher than the OCR of cells treated with HC sera. To test this hypothesis, we followed the methods described in Fluge et al (2016). In our pre-registered analysis plan we defined our primary outcome as the OCR under conditions of maximal respiratory capacity, measured after the injection of FCCP into the wells. Our secondary outcomes were differences in OCR and ECAR under the other measured conditions: basal (amino acids), basal+glucose, and proton leak (https://osf.io/qwp4v, 02/08/2024).

On each 96-well plate, we treated healthy myoblasts with sera from up to 18 randomised and blinded participants (a "sample set", e.g., A,B,C) and performed 5 technical replicates per participant per plate. Each sample set was tested on 3 plates (e.g., plates AI, AII, AIII), yielding 15 technical replicates per participant.

The changes in OCR (Fig 2A) and ECAR (Fig 2B) followed expected patterns under the different conditions, shown here from one plate, BII. Across all 22 plates and 1,926 measured wells, there was no significant difference in average cell counts between the two groups (Fig 2C). Representative images from three wells treated with ME serum and three wells treated with healthy control serum illustrate that the cells remained intact throughout the assay (Fig S2 in S1 File). OCR and ECAR are both dependent on cell numbers, and normalizing measures based on cell numbers was carried out as recommended by Agilent [24,25]. Indeed, we observed significant positive correlations between cell count and measurements at maximal respiratory capacity for both OCR ($r^2 = 0.36$, $p < 0.0001$) and ECAR ($r^2 = 0.88$, $p < 0.0001$) (Fig 2D, 2E). Consequently, we corrected for differences in cell counts in all subsequent analyses.

## Technical sources of variation

The Seahorse mito stress test assay is known to be prone to technical variability [24]. Prior to unblinding, we examined the effects of cell count on maximal respiratory capacity, and the plate-to-plate variation. Technical sources of variation were evident as plate effects, explaining 42% of the total variance, while sample position ("well effects") contributed only 4% of the total variance. Repeatability between technical replicates for OCR at maximal respiratory capacity for a given participant was estimated to be 0.56. Given the length of the exposure in cell culture, the known variability of the Seahorse assay, and the lack of group differences, we consider this to reflect good technical repeatability. Technical variability is expected within such a large experiment. However, our experimental design, with cases and controls randomised and present on each plate, ensured that differential biological effects of individuals' sera on cells should have been captured had they been present.

## Primary outcome: No difference in maximal respiratory capacity

For our primary analysis, we asked whether OCR at maximal respiratory capacity was higher in myoblasts that had been treated with ME patient sera than with healthy control sera. Maximal respiratory capacity averaged across technical replicates for each participant was similar between the two groups (Fig 3A, 3B). Despite substantial plate-to-plate variation of OCR at maximal respiratory capacity, the ME and HC groups did not differ within any particular plate (Fig 3C). We analysed our data with a mixed effects model that corrected for the correlation between cell counts and OCR, and accounted for the differences between plates, model 1:

$$maximal\_respiratory\_capacity \sim group + scale(cell\_counts) + (1 \mid sample\_ID) + (1 \mid plate\_id)$$

This model was applied to provide the best chance of observing changes due to biological differences rather than to technical artefacts.

The primary analysis yielded a clear null result. Serum from pwME or healthy controls did not differentially affect OCR at maximal respiratory capacity, with the ME group effect estimated at 2.50 pmol/min higher than controls, yet with its 95%

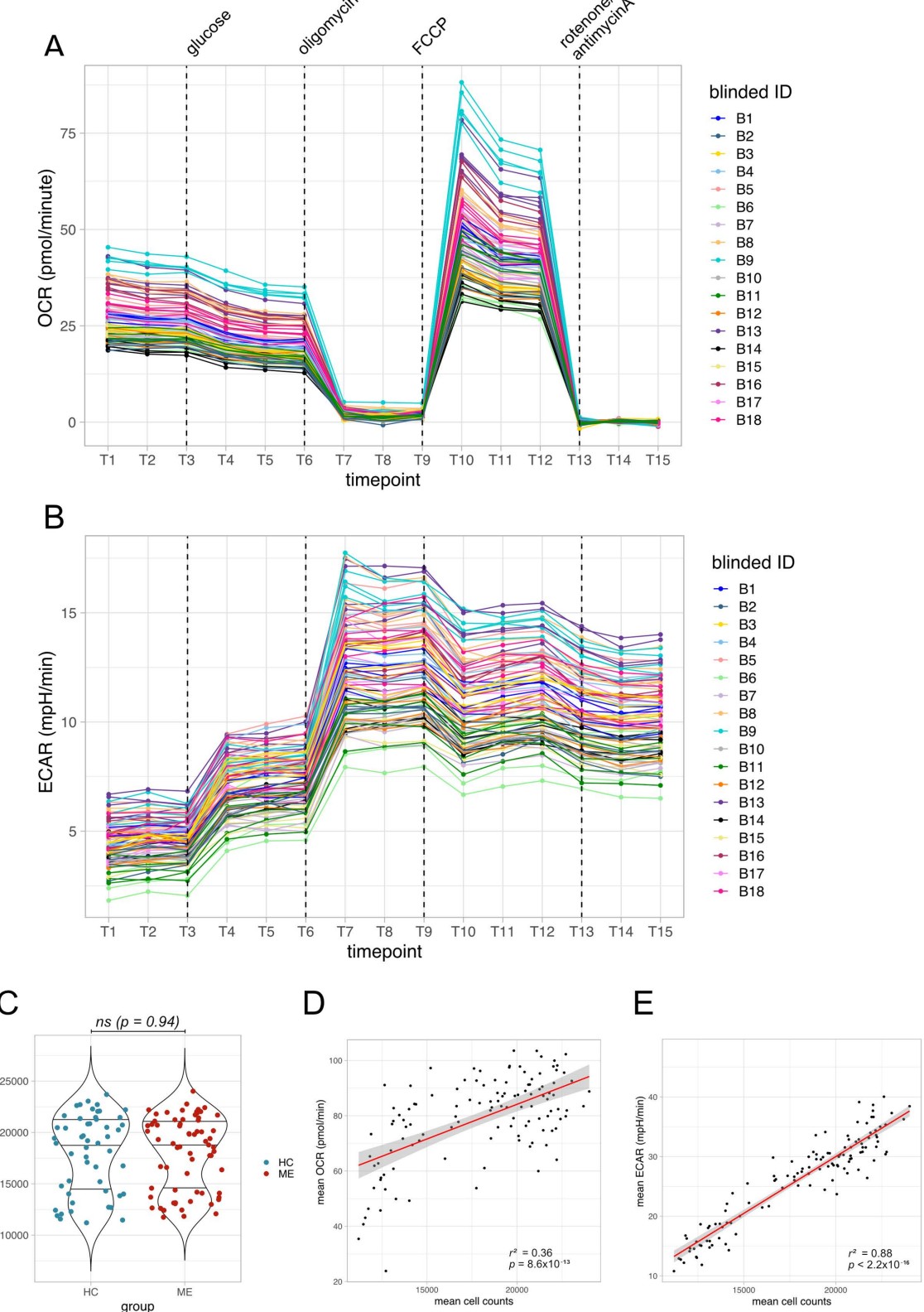

**Fig 2. Validating assay performance.** (A) Data for oxygen consumption rate (OCR) for all wells on a single plate, BII, across all 15 timepoints when measurements were taken, and coloured by blinded individual ID. Dashed lines indicate when substrates and drugs were added. **(B)** Same conditions

but showing ECAR measurements on plate BII. **(C)** Averaged cell counts from wells treated with each individual's serum, shown by group. Differences in cell count between groups were tested with a Wilcoxon Rank-Sum Test, alpha = 0.05; ns = not significant. **(D)** Correlations between mean OCR and mean cell counts, averaged across all measurements for each participant, and (E) mean ECAR and cell counts at maximal respiratory capacity. Annotated with squared Pearson correlation coefficients and p values.

confidence interval including zero (−1.62 to 6.62); the associated p value lay above the significance threshold, p = 0.23. To aid in visualising the results, we plotted the residuals from model 0 defined as:

$$maximal\_respiratory\_capacity \sim scale(cell\_counts) + (1 \mid plate\_id)$$

Model 0 corrects for cell count and plate effects only, allowing us to visualise any variation in the data that is due to group differences between HC and ME. Averaged residuals from model 0 showed substantial overlap in maximal respiratory capacity between the two groups (Fig 3B). While substantial variation in OCR at maximal respiratory capacity was observed between individual sera, no systemic differences occurred between the two groups (Fig 3D).

### Secondary outcomes: OCR under other conditions, and Extracellular Acidification Rate (ECAR) do not differ between cells treated with ME sera or healthy control sera

ME status did not affect OCR for measurements taken under the 3 other conditions: basal amino acids (Fig 3E), basal glucose (Fig 3F), or proton leak (Fig 3G). Furthermore, there were no significant effects of ME sera on ECAR at maximal respiratory capacity (Fig 4A), as shown by Model 0 residuals for ECAR (Fig 4B). When analysed with the model 1 predictors, ECAR in the ME sera treated group and the healthy control sera treated group were not significantly different (estimate: −0.61, 95% CI [−1.75,0.53], p = 0.29). No differences were observed under the other conditions: basal (amino acids) (Fig 4C), basal (+ glucose) (Fig 4D), or proton leak (Fig 4E).

### Sensitivity analyses: Maximal respiratory capacity by disease severity, age, BMI, and sampling batch

We hypothesised that cohort characteristics or batch effects could have masked effects of ME serum exposure on OCR. Stratifying cases by severity demonstrated no severity-dependent effects on maximal respiratory capacity (Fig 5A). Since increased BMI is associated with changes in the levels of hormones such as leptin [26], and metabolites (including triglycerides and glucose) in the blood [27], we hypothesised that participant BMI could affect the OCR of cultured myoblasts. Given the small but statistically significant difference in BMI between the two cohorts, we calculated the correlation between BMI and OCR at maximal respiratory capacity, yet this failed to reach statistical significance (Fig 5B). To assess the robustness of our model's results, we added BMI as a predictor to model 1. Nevertheless, this failed to affect the outcome (estimate for ME group: 1.73, 95% CI [−4.19, 7.65], p = 0.56). Age also did not correlate with OCR at maximal respiratory capacity (Fig 5C). We hypothesised that the differences in storage time between the two sampling batches collected at different time periods could have introduced technical bias into the data, since pwME and controls were not matched 1:1 in the sampling batches. Batch effects were tested by adding *batch* as a predictor to model 1 and examining whether the estimate for *batch* was significant:

$$maximal\_respiratory\_capacity \sim group + scale(cell\_counts) + batch + (1 \mid sample\_ID) + (1 \mid plate\_id)$$

A greater dispersion of values from batch 2 was observed which could be due to the shorter storage time compared to batch 1, donor variability, or other batch effects (Fig 5D). However, no significant effect of *batch* on OCR at maximal respiratory capacity was observed: estimate for *batch*, −0.81, 95% CI [−5.56, 3.93], p = 0.73. Similarly, neither BMI (Fig 5E) nor age (Fig 5F) was correlated with ECAR at maximal respiratory capacity, and there was no significant effect of batch

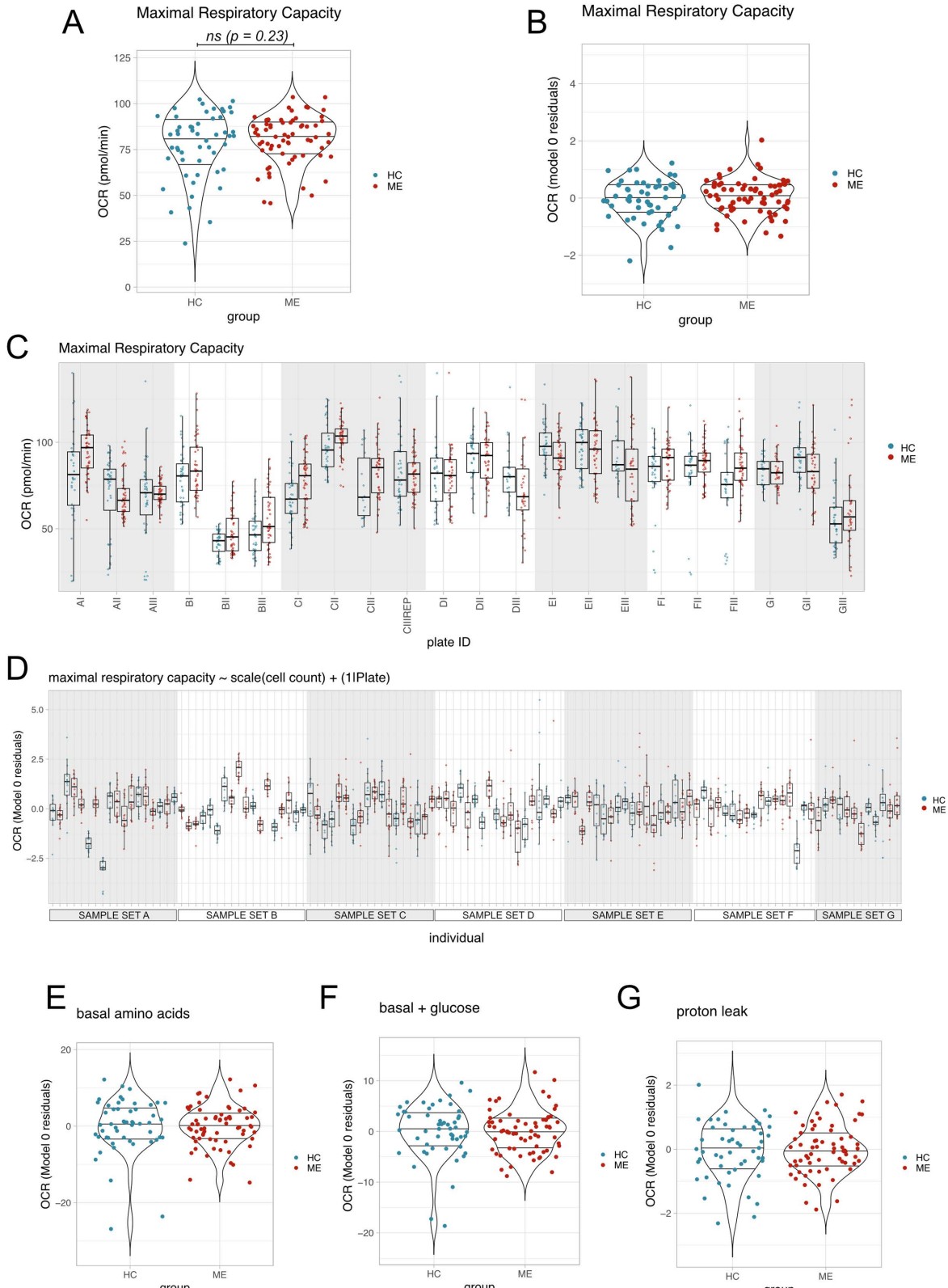

**Fig 3. OCR at maximal respiratory capacity and under other conditions. (A)** OCR at maximal respiratory capacity averaged across technical replicates for each participant. Group differences were tested using model 1, ns = not significant. **(B)** Residuals, averaged by participant, for OCR at maximal

respiratory capacity from model 0. **(C)** All raw OCR measurements for each well in each plate at maximal respiratory capacity. Plate CIII was repeated "CIII rep". **(D)** Residuals from model 0 shown for each participant across all technical replicates, grouped by sample set. **(E)** Mean model 0 residuals for OCR measurements taken under conditions: basal (amino acids), **(F)** basal, after glucose addition, or **(G)** proton leak.

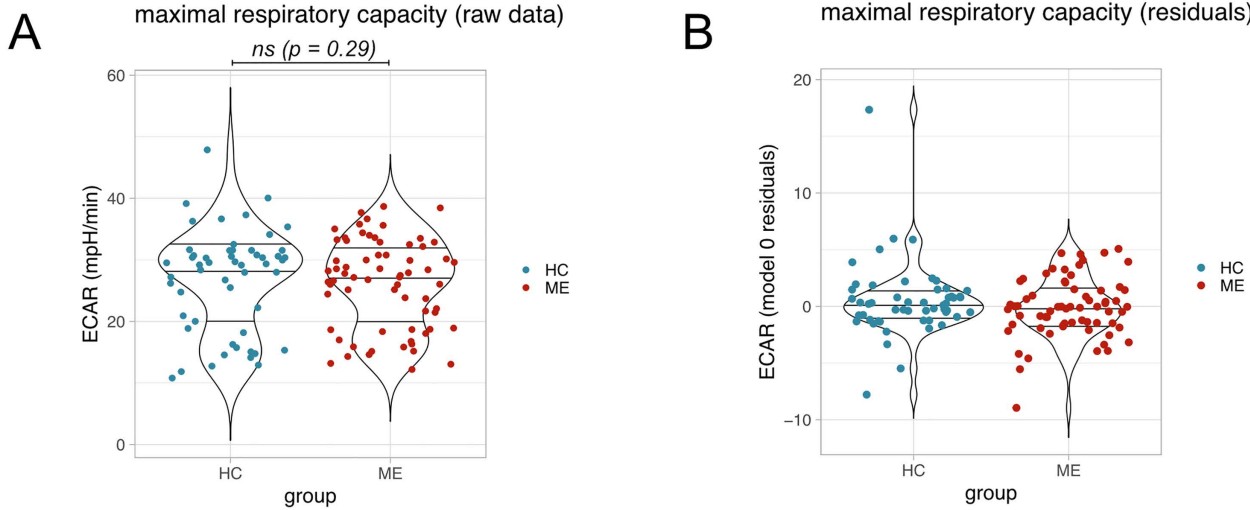

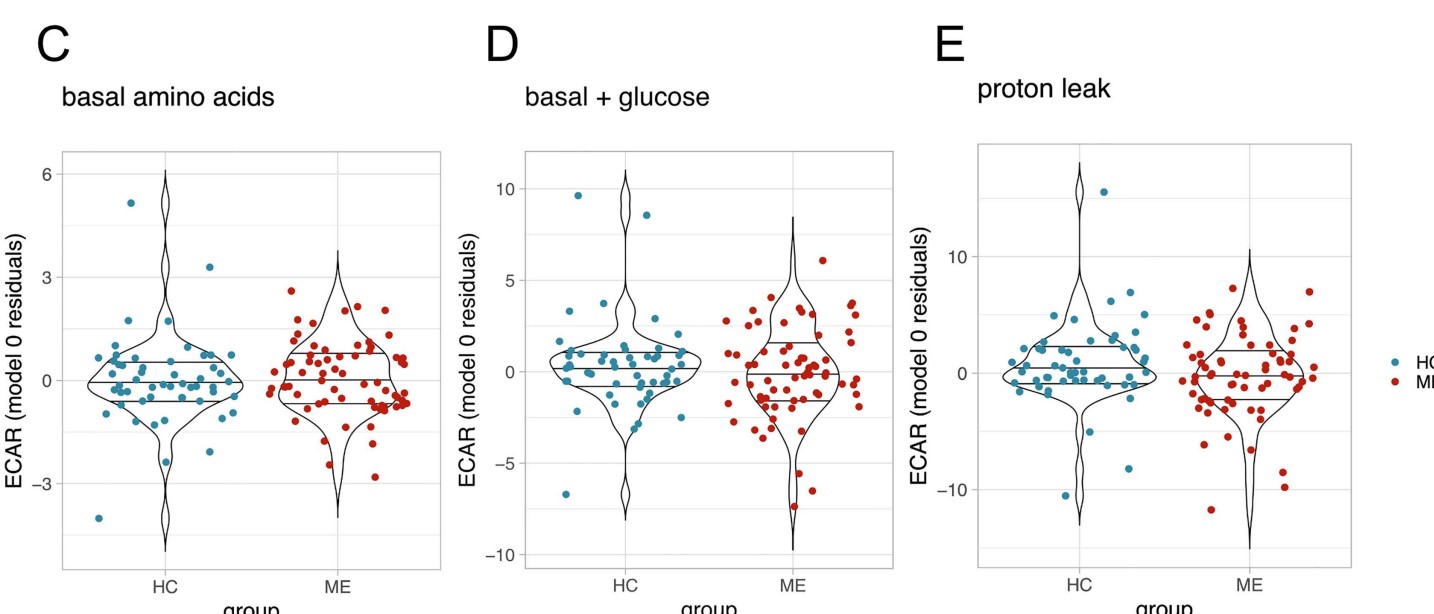

**Fig 4. ECAR under tested conditions. (A)** ECAR at maximal respiratory capacity averaged across all technical replicates for each participant. Group differences were tested using model 1, ns = not significant. **(B)** Residuals, averaged by participant, for ECAR at maximal respiratory capacity from model 0. **(C)** Averaged ECAR model 0 residuals for measurements taken under conditions: basal (amino acids), (D) after glucose addition, or (E) proton leak.

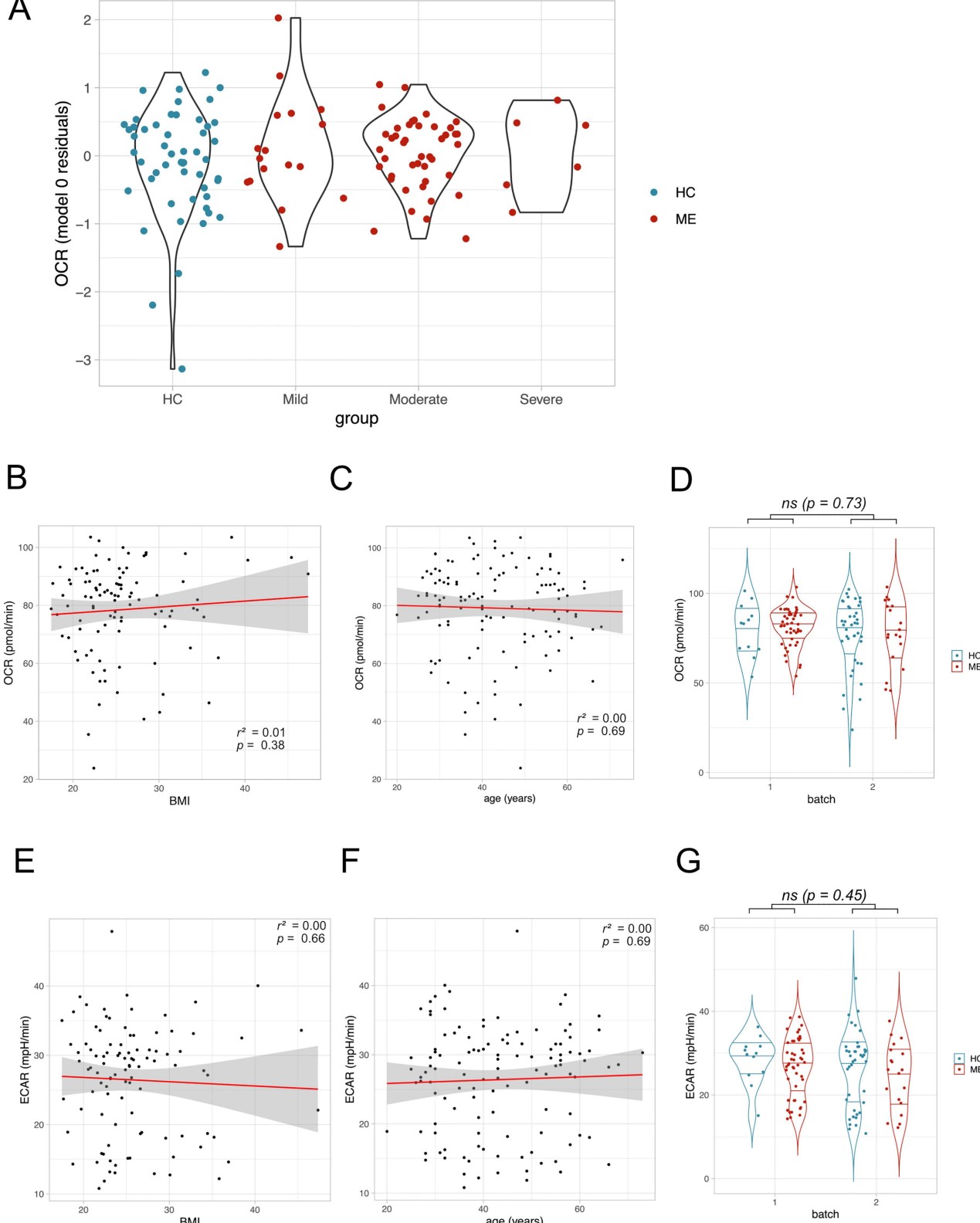

**Fig 5. Sensitivity analyses for severity, age, BMI and sampling batch. (A)** OCR residuals at maximal respiratory capacity averaged by individual, and stratified by severity. **(B)** OCR at maximal respiratory capacity averaged by individual and correlated with BMI and **(C)** age, and tested using

Pearson's correlation coefficient at significance level alpha = 0.05. **(D)** OCR at maximal respiratory capacity averaged by individual. **(E)** ECAR at maximal respiratory capacity averaged by individual and correlated with BMI and **(F)** age, and tested using Pearson's correlation coefficient at significance level alpha = 0.05. **(G)** ECAR at maximal respiratory capacity averaged by individual. Batch effect was tested as before but using ECAR as the response variable instead of OCR. ns = not significant.

on ECAR at maximal respiratory capacity (Fig 5G; estimate for batch −0.51, 95% CI [−1.83, 0.81], p = 0.45). Since ME is likely to be a heterogeneous disease [28], we performed a post-hoc analysis examining whether stratifying the ME cohort based on other disease characteristics might reveal ME subgroups for which serum exposure affected OCR (Fig S3 in S1 File). Stratifying the ME cohort based on disease duration, trigger types, and illness course did not provide any evidence for subgroup-specific serum effects on OCR. Thus, none of the cohort characteristics and technical variables we considered had an impact on the OCR and ECAR measurements taken at maximal respiratory capacity.

## Discussion

Despite our best efforts to replicate the study and findings from Fluge et al (2016), these results failed to demonstrate an effect of ME serum on increased OCR in cultured myoblasts. Our pre-registered replication study was performed on a well-defined cohort of 67 pwME and 53 healthy controls, and its design ensured that sources of technical variation could be accounted for. Our study performed 1,926 stress tests under blinded experimental conditions, with pre-defined outcome measures. We saw no difference in our primary outcome (difference in OCR at maximal respiratory capacity), nor in OCR and ECAR under any other of the conditions we measured. Consequently, our study's results do not support the hypothesis that ME sera impact on healthy myoblast mitochondrial phenotypes differently from healthy control sera.

The effect size of our outcome of interest (OCR at maximal respiratory capacity) in the original study was estimated as 1.32 [95% CI 0.43–2.2] (Cohen's D). In replication studies, however, effect sizes are expected to be reduced [29]. We modelled the statistical power of our study based on the sample sizes of patient and control groups, assuming different effect sizes, and determined that with an effect size as low as 0.52 (less than half the original effect size), we expected to achieve 80% power to detect a statistically significant difference at a significance threshold of 0.05.

In our pre-registration we reported the following differences between our study and that of Fluge et al (2016), summarised in Table 2:

**Table 2. Differences between Fluge et al (2016) and our replication study.**

| Feature | This study | Fluge et al 2016 |
|---|---|---|
| Cell line: HSMM Lonza | Donor female, BMI ~ 17, age 31, lot number 21TL138913 | Unknown healthy donor |
| Drugs | 2 µM Carbonyl cyanide-p-trifluoromethoxyphenylhydrazone (FCCP), 2 µM Oligomycin | 2 µM Carbonyl cyanide m-chlorophenyl hydrazone (CCCP), 3 µM Oligomycin |
| Culture plate | Seahorse Pro M plates (with moat to reduce edge effects) | Unknown, no moat although did not use wells along edges |
| Normalisation | By cell count | Data not normalised |
| Patient characteristics | Mostly moderate/mild ME | Severe/very severe ME |
| Diagnostic criteria | Canadian Consensus Criteria **and/or** Institute of Medicine criteria + self-reported medical diagnosis by a healthcare professional | Canadian Consensus Criteria |

1) Although we used the same supplier to procure human skeletal muscle myoblasts and culture media, the cells are likely to be derived from different donors (Table 2). While the lot number from Fluge et al. was not available to confirm this, it is likely our cells came from different donors, since the studies were performed nearly 10 years apart. The donor had a low BMI, however, given these myoblasts are sold for commercial use in cell culture assays we expect that they behave comparably to other myoblasts available from this supplier. While genetic differences in the HSMM cells could have altered their susceptibility to ME serum, it is reasonable to expect that the assay would be generalisable to other healthy HSMMs beyond those used in the original study.

2) Mitochondrial uncoupler drugs used for our study (FCCP) and Fluge et al.'s (CCCP) are validated for the Mito-Stress Test and have the same mechanism of action. We chose FCCP because it is recommended by the manufacturer and is an industry standard for the mitochondrial stress test [30]. We found oligomycin to produce equivalent results at 2 µM and at 3 µM (Fig S4 in S1 File) and again used 2 µM which is the maximum recommended concentration by Agilent.

3) The cell culture plates used in our study are the industry-standard recommended cell culture plates "Seahorse Pro M plates". These include a moat along the edge of the plate which is filled with sterile water. This reduces evaporation in the wells adjacent to the edge of the plates. Additionally, to address any potential well-effects, we ensured that each sample was present in edge-wells in only 1 of the 3 plate replicates. When well-effects on OCR were estimated, they only explained 4% of the variance, which we considered negligible.

4) In Fluge et al, data were not normalised by cell count or protein concentration. We observed a strong correlation of OCR and ECAR with cell counts and therefore accounted for cell numbers in our analysis. However, no differences in cell counts were observed between ME and HC, so are unlikely to have contributed to ME serum effects previously observed by Fluge et al.

5) Samples were obtained from people with severe and very severe ME in Fluge et al. A majority of our samples came from people with moderate ME. However, the analysis stratified by severity does not indicate increased maximal respiratory capacity in individuals with severe ME in our cohort, or a correlation of maximal respiratory capacity with severity. Nonetheless, if the effect of serum on myoblasts were specific to people with severe or very severe ME, it is possible that with the small number of people with severe ME in our study we were under-powered to replicate that result.

Future studies of ME would benefit from standardised cohort characterisation to facilitate replication and direct comparison between research findings from different cohorts.

A further limitation of our study is that participants with ME may not have been experiencing post exertional malaise (PEM) on the day of sampling. Participants had to travel to the university site to donate a blood sample, and due to ethical concerns around inducing crashes, we encouraged participant to re-schedule if they were not able to attend the site on that day. If ME-biased factors are episodic in people with mild and moderate ME, and only present when they experience PEM, in contrast to people with severe ME where they are present all the time, it could explain the difference in findings between this study and Fluge et al (2016). Since many pwME experience a fluctuating illness course, future studies of blood factors should consider sampling individuals longitudinally on days when participants are experiencing PEM, and days when they are not, to maximise their likelihood of capturing PEM-related biomarkers.

Our results do not rule out the possibility of ME-biased factors being present in serum, but they do not support the use of this experimental method for detecting such factors. For example, a recent study profiling cell free RNA in plasma identified 743 unique features that differed between ME cases and controls particularly related to platelets, plasmacytoid dendritic cells, monocytes, T cells and potential dysregulated mtRNA expression [31]. Furthermore, we cannot rule out the occurrence of other molecular adaptations in the blood or in the myoblasts such as compensatory mechanisms that could rescue effects of factors in the blood on myoblasts. Such adaptations could be detected by measuring changes in gene expression or by proteomics. Future studies in which cell cultures are exposed to ME or healthy serum longitudinally could

determine whether temporal changes and adaptations occur that may have been missed in our study. Finally, DecodeME, a genome-wide association study of ME, identified a candidate gene (FBXL4) involved in mitophagy and mitochondrial DNA depletion [28]. This suggests that mitochondrial dysfunction may well be relevant to ME pathogenesis, but that healthy myoblasts might not have the relevant genetic susceptibilities to produce altered metabolic phenotypes. Future studies examining the role of FBXL4 in ME will help clarify the role of mitochondrial dysfunction in ME.

Given the large sample size in our study and the large number of technical replicates we performed, minor differences in the cell lines or assay conditions are unlikely to have masked an ME-biased biological effect of the serum on the myoblasts. We consider our study to provide strong evidence that ME serum biased effects on healthy myoblast mitochondrial phenotypes are not generalisable. Future studies may benefit from exploring compartments other than blood for the discovery of disease-specific factors. This study cautions against the translational relevance of previous evidence of ME serum factors altering mitochondrial phenotypes in healthy cultured cells and demonstrates the importance of replicating ME research findings with well-powered sample sizes.

## Supporting information

**S1 File. Supplementary Figures.**
(DOCX)

**S2 File. OCR measurements (rotenone adjusted).**
(CSV)

**S3 File. ECAR measurements.**
(CSV)

**S4 File. Cohort characteristics.**
(CSV)

**S5 File. R analysis scripts.**
(R)

## Acknowledgments

The Seahorse Mito Stress Tests were carried out by the EdinOmics research facility (RRID: SCR_021838) at the University of Edinburgh. This research was conducted with the assistance of the Edinburgh Genome Foundry, an engineering biology research facility specialising in the modular, automated assembly of DNA constructs and phenotypic characterisation at the University of Edinburgh. We are grateful to the phlebotomy team at SHU who helped with this study, to the Sheffield ME and Fibromyalgia Group, and to all the participants in this study. This study was carried out with the help of a patient and public involvement panel who provided input at all stages of the project. Specifically, we would like to acknowledge Maree Candish, Anja Demmel, Clare Rachwal and Simon McGrath for their contributions to this study.

## Author contributions

**Conceptualization:** Audrey A. Ryback.

**Data curation:** Audrey A. Ryback, Camila M. Loureiro, Caroline F. Dalton.

**Formal analysis:** Audrey A. Ryback, Charles B. Hillier.

**Funding acquisition:** Audrey A. Ryback.

**Investigation:** Audrey A. Ryback.

**Methodology:** Audrey A. Ryback, Charles B. Hillier, Chris P. Ponting, Caroline F. Dalton.

**Project administration:** Audrey A. Ryback.

**Resources:** Caroline F. Dalton.

**Writing – original draft:** Audrey A. Ryback, Charles B. Hillier.

**Writing – review & editing:** Audrey A. Ryback, Camila M. Loureiro, Chris P. Ponting, Caroline F. Dalton.

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
