## [Editor Report · Decision Letter 0]

13 Aug 2025

Dear Dr. Ryback,

We look forward to receiving your revised manuscript.

Kind regards,

Sadiq Umar

Academic Editor

PLOS ONE

Additional Editor Comments:

To enhance the mechanistic depth of the study, I strongly recommend assessing gene expression for key glycolysis and oxidative phosphorylation (OXPHOS) pathway components. This will clarify whether metabolic reprogramming is occurring despite the indistinguishable mitochondrial phenotypes.

Additionally, examine markers of cellular exhaustion/metabolic stress signaling, similar to the approach in PMID: 40789036. Such analysis could uncover compensatory or maladaptive bioenergetic responses that may not be evident from mitochondrial functional readouts alone.

Including these analyses will substantially strengthen the conclusions and improve the translational relevance of the findings.

---

## [Author Response · Author response to Decision Letter 1]

29 Sep 2025

Dear Editor,

Thank you for taking the time to consider our manuscript and for your helpful comments and feedback.

In light of your review, we have revised key sections in the Introduction and Discussion sections to better contextualise our results within the body of ME literature and to highlight limitations of examining cellular rather than molecular phenotypes. We have expanded on what future studies might explore, and further clarified the contribution that our well-powered study has made to an evidence base that often lacks rigour and reproducibility.

We here address each of your 3 points in turn:

1) Enhancing the mechanistic depth of the study by assessing gene expression for glycolysis/OXPHOS components.

We agree with the value of performing additional experimental work to explore possible mechanisms of ME serum-induced mitochondrial phenotypes beyond the cellular phenotypes reported in our study. The limited funding and sample material available to us, however, makes this additional work impossible. The cost of performing RNA sequencing or qPCR of key glycolysis and OXPHOS genes on 120 cell cultures exposed to ME or control serum far exceeds the remaining budget on this charity-funded grant.

We agree that had we observed any differences in cellular phenotype between our disease and control groups then this would have been a particularly informative additional work package. It is possible that metabolic reprogramming is occurring and that changes at a molecular level might be observed in the myoblasts with methods such as gene expression quantification. However, this question lay beyond the scope of our pre-registered study which was focussed, instead, on changes in phenotype at the cellular level. You have raised an important point and hence we have now commented on this limitation of our work in the Discussion section.

2) Compensatory or maladaptive metabolic responses measured by orthogonal assays such as studying cell free RNA.

Examining the possibility of compensatory/maladaptive metabolic responses in ME is certainly of interest to the field. Our assay, however, was intended instead as a study system that could screen for differences in serum factors between ME and controls, rather than examining mechanisms of metabolic dysfunction in ME directly. The study by Gardella et al 2025 (https://doi.org/10.1073/pnas.2507345122) that you pointed to (which, notably, was published after we submitted our paper to your journal - 11/08/2025), provides an interesting hypothesis-generating approach to identify novel biomarkers for ME using cell free RNA. Their findings identify that certain cell free RNA transcripts – including a significantly lower percentage of mtRNA in those with ME - might be used to distinguish healthy control and pwME plasma using machine learning. While their results suggest that metabolic dysfunction may be observed in the blood from people with ME, we do not expect cell free RNA in the blood, which originates mainly from dying platelets, red blood cells and leukocytes, to be directly relevant to the metabolic adaptations of the cultured myoblasts exposed to ME serum in our study. Unfortunately, performing cell free RNA quantification is again not feasible due to our limited remaining research budget and available serum, and because studies exploring cell free RNA have demonstrated low reproducibility and high technical and biological variation (https://doi.org/10.1186/s40364-022-00409-w).

3) Strengthening the conclusions and improving the translational relevance of the findings.

This work demonstrates that using in vitro models of examining cellular phenotypes of healthy myoblasts exposed to serum from people with ME is unlikely to lead to translational findings. This knowledge is critical for the ME field in which translational research remains in its infancy. Establishing a firm evidence base is crucial and without the publication of null results and replication studies, future research will continue to follow fruitless lines of enquiry.

Finally, we wanted to emphasise why we believe our paper merits publication in PLOS One and fits the scope of your journal. The PLOS One website states: “We evaluate research on the basis of scientific validity, strong methodology, and high ethical standards—not perceived significance. Multidisciplinary and interdisciplinary research, replication studies, negative and null results are all in scope. We also publish Registered Reports and Protocols” (https://journals.plos.org/plosone/s/journal-information). Unlike many other journals that are concerned only with the perceived novelty of results, PLOS One’s emphasis on rigour and good scientific practice regardless of negative results is a key reason why we chose to submit our work to your journal. It is our conviction that the scientific rigour with which our research was carried out, including preregistration of our study on the Open Science Foundation (https://osf.io/qwp4v), makes it well-suited to the scope of PLOS One, despite the negative findings.

Thank you for considering our manuscript and we look forward to hearing from you soon.

With very best wishes,

Audrey Ryback

---

## [Decision Letter · Decision Letter 1]

3 Nov 2025

Dear Dr. Ryback,

Thank you for submitting your manuscript to PLOS ONE. After careful consideration, we feel that it has merit but does not fully meet PLOS ONE’s publication criteria as it currently stands. Therefore, we invite you to submit a revised version of the manuscript that addresses the points raised during the review process.

We look forward to receiving your revised manuscript.

Kind regards,

Sadiq Umar

Academic Editor

PLOS ONE

Journal Requirements:

Reviewers' comments:

Reviewer's Responses to Questions

**Comments to the Author**

Reviewer #1: (No Response)

Reviewer #2: (No Response)

Reviewer #3: All comments have been addressed

2. Is the manuscript technically sound, and do the data support the conclusions?

Reviewer #1: Yes

Reviewer #2: Yes

Reviewer #3: Yes

3. Has the statistical analysis been performed appropriately and rigorously?

Reviewer #1: I Don't Know

Reviewer #2: Yes

Reviewer #3: Yes

4. Have the authors made all data underlying the findings in their manuscript fully available?

Reviewer #1: Yes

Reviewer #2: Yes

Reviewer #3: Yes

5. Is the manuscript presented in an intelligible fashion and written in standard English?

Reviewer #1: Yes

Reviewer #2: Yes

Reviewer #3: Yes

Reviewer #1: This study aimed to replicate findings from Fluge et al. (2016), which suggested that serum from individuals with Myalgic Encephalomyelitis/Chronic Fatigue Syndrome (ME/CFS) could increase mitochondrial respiratory capacity in healthy myoblasts. Using a larger sample size (67 ME/CFS patients and 53 healthy controls), researchers treated cultured myoblasts with serum and conducted over 1,700 mitochondrial stress tests using a Seahorse Bioanalyser. Contrary to the original findings, they found no significant differences in oxygen consumption rates at maximal respiratory capacity between the ME/CFS and control groups. These results challenge the hypothesis that ME/CFS serum contains factors that alter mitochondrial function in vitro, suggesting limited utility for this approach in developing diagnostic tests.

Strengths: The study is commendable for its careful experimental design, robust methodology, and transparent discussion of potential biases and limitations. Additionally, the authors provide a clear rationale for their approach and acknowledge the constraints of their experimental system.

Suggestions for Improvement: It would have been beneficial to include microscopic monitoring of the assays, with representative cell images and viability dye staining/counts before and after treatment. This would help assess potential morphological changes or cytotoxic effects that might not be captured by metabolic measurements alone.

Weaknesses and Considerations: A key limitation lies in the heterogeneity of the ME/CFS patient population, which may obscure subtle serum-induced effects. The wide variability in OCR and ECAR values across individual samples further complicates interpretation. Although the study increased the sample size compared to Fluge et al. (2016), the low representation of severely affected patients may limit the generalizability of the findings, which appear restricted to mild/moderate cases. While stratification by severity did not yield significant differences, it would be valuable to know whether the authors conducted post hoc stratified analyses based on other clinical or biological traits or employed dimensionality reduction techniques such as PCA to explore latent patterns.

Data Presentation: In Figures 4D and 4G, the greater dispersion of values in batch 2 for both study groups is noticeable. A brief commentary on potential causes (such as batch effects, sample handling, or donor variability) would help readers better understand this discrepancy and its implications for data interpretation.

In Figure 4, it appears that 1–2 data points in the healthy control (HC) group may be outliers. It would be helpful if the authors addressed this, either by discussing their impact or clarifying whether any statistical treatment was applied.

Tables 1 through 3 could be consolidated into a single comprehensive table to improve readability and reduce redundancy.

Figures should include exact p-values, in addition to the current "n.s." (not significant) labels. Consistency in formatting across all figure labels would enhance clarity.

There is a typographical error on line 203.

Reviewer #2: The manuscript presents a well-designed, preregistered replication of the study by Fluge et al. (2016), evaluating the effects of ME/CFS patient serum on mitochondrial function in cultured myoblasts. The study is technically solid, uses a sufficiently powered sample, and applies rigorous statistical methods with proper control of plate effects and randomization. The authors should be commended for their transparency, detailed reporting, and for publishing negative results—an important and often underrepresented aspect of the scientific process. These features make the paper a valuable contribution to the ME/CFS field, which greatly benefits from efforts to test reproducibility under controlled conditions.

However, while the experimental rigor is strong, the narrative emphasis of the paper leans heavily toward discrediting the results of Fluge et al., rather than leveraging this replication outcome to address broader questions about biological variability and cohort standardization in ME/CFS research. A more balanced framing would increase the paper’s constructive impact and relevance beyond a single prior study.

To be specific, the study convincingly shows that no serum-driven mitochondrial phenotype was observed under the specific experimental and cohort conditions tested here. Yet, direct comparability with Fluge et al. remains limited due to several biological and clinical differences between cohorts:

(1) Sex distribution: This study includes only women, while Fluge et al. analyzed both sexes and observed stronger effects in females.

(2) Disease severity and duration: The original cohort contained mostly moderate-to-severe, long-standing ME/CFS cases, while the present cohort is milder, and disease duration is not clearly reported.

(3) Age, medication, fasting state, and comorbidities: These variables are not fully described and may affect circulating metabolites and mitochondrial behavior.

These distinctions suggest that both studies may not be addressing precisely the same biological question, even if the technical protocol is similar. Therefore, it would be more accurate to interpret the current results as indicating no detectable effect within this specific cohort, rather than as a categorical refutation of previous findings.

I encourage the authors to revise the Discussion to highlight this key point and to extract a broader methodological message for the field: the necessity of standardized and well-documented cohort characterization (sex, age, disease duration, severity, metabolic state, and preanalytical conditions). Such harmonization would greatly enhance reproducibility and interpretability across future ME/CFS studies.

In summary, this manuscript has clear value due to its methodological strength, transparency, and negative results. Reframing the discussion from a refutation to a constructive call for methodological standardization would strengthen the scientific and conceptual contribution of this work.

Reviewer #3: Introduction

You have introduced the term “people with56 ME (pwME).” Please not that The acronym “pwME” is not universally recognized and may confuse readers unfamiliar with the convention. It is not universal to describe people suffering from other conditions (heart failure etc.) as people with XX. Using such approach might further distant the general public view of the field of ME/CFS from rest of biologically-based severe conditions.

Methods

“Sera from pwME and HC were collected between 27/11/2023 - 23/02/2024 across two117

rounds of sampling over two weeks in November–December 2023 (“batch 1”), and118

three weeks in February 2024 (“batch 2”) (Table 1).” Do You think that it might affect results obtained? Please describe it as a potential limitation.

“Due to the female preponderance119

of ME (3) and to reduce heterogeneity, all study participants were female.

“ Please describe it as a potential limiting factor of the study

“People with ME met the121

Canadian Consensus Criteria (CCC) and/or the Institute of Medicine (IoM) diagnostic122

criteria and reported a diagnosis of ME by a healthcare professional. Healthy controls123

did not meet the CCC or IoM criteria according to their screening survey responses124

and did not report any of the 21 active comorbidities screened for by the DecodeME125

screening questionnaire (13).” How many met IOM and how many met CCC? How ME patients and HCs were recruited? Please describe, using flowchart might help

Discussion:

“Consequently, our study’s results do not support the hypothesis that442

ME sera impact on healthy myoblast mitochondrial phenotypes differently from healthy443

control sera.” Can You cite previous studies on blood cell‐based diagnostic test in CFS, that would effectively delineate patients vs controls? What variables were taken into account in those studies?

“A further limitation of our study is that participants with ME may not have been501

experiencing post exertional malaise (PEM) on the day of sampling. “ Can You describe briefly previous studies on how physical activity affects mitochondria in CFS patients?

The cohort includes mostly mild-to-moderate ME/CFS cases, whereas Fluge et al. studied severe/very severe patients. While the authors acknowledge this, they could more explicitly discuss whether their null result might be due to disease heterogeneity rather than a true refutation of the original finding. Please add a paragraph in the Discussion exploring whether ME/CFS subtypes (e.g., based on PEM severity, onset type, or immune profile) might differentially affect serum bioactivity. Cite recent work (e.g., from DecodeME or NIH intramural studies) suggesting ME/CFS is not a monolithic condition.

The authors note that participants were not necessarily experiencing post-exertional malaise (PEM) at blood draw, which could mask transient serum factors. What kind of a future study design where blood is collected before and after a stressor could be done? What limitations such study would have, including ethical concerns?

Please discuss whether alternative assay platforms might be better suited for detecting small effect sizes in future work.

In the limitations or future directions, mention that normalization could provide additional resolution, especially if serum factors alter mitochondrial density without changing per-mitochondrion function.

**Do you want your identity to be public for this peer review?** For information about this choice, including consent withdrawal, please see our Privacy Policy

Reviewer #1: No

Reviewer #2: No

Reviewer #3: No

---

## [Author Response · Author response to Decision Letter 2]

22 Dec 2025

We would like to thank all three reviewers for their constructive feedback and suggestions, and have addressed their points in turn. Please note that line numbers referred to in this document correspond to the line numbers in the version of the manuscript with tracked changes.

Reviewer #1:

"This study aimed to replicate findings from Fluge et al. (2016), which suggested that serum from individuals with Myalgic Encephalomyelitis/Chronic Fatigue Syndrome (ME/CFS) could increase mitochondrial respiratory capacity in healthy myoblasts. Using a larger sample size (67 ME/CFS patients and 53 healthy controls), researchers treated cultured myoblasts with serum and conducted over 1,700 mitochondrial stress tests using a Seahorse Bioanalyser. Contrary to the original findings, they found no significant differences in oxygen consumption rates at maximal respiratory capacity between the ME/CFS and control groups. These results challenge the hypothesis that ME/CFS serum contains factors that alter mitochondrial function in vitro, suggesting limited utility for this approach in developing diagnostic tests.

Strengths: The study is commendable for its careful experimental design, robust methodology, and transparent discussion of potential biases and limitations. Additionally, the authors provide a clear rationale for their approach and acknowledge the constraints of their experimental system.

Suggestions for Improvement: It would have been beneficial to include microscopic monitoring of the assays, with representative cell images and viability dye staining/counts before and after treatment. This would help assess potential morphological changes or cytotoxic effects that might not be captured by metabolic measurements alone."

We agree that microscopic monitoring of cells in response to serum treatment would be an interesting avenue of investigation. Showing morphological changes to the cells that would be statistically representative over the complete course of the experiment lay beyond the scope of this study’s aims. In the revised submission we include Brightfield and Hoechst-stained images of cells after treatment to illustrate that cells remained intact: see additional Figure: Supplementary Figure 2, discussed in the manuscript (see lines: 294-296-highlighted in yellow). We agree that it was important that we accounted for potential cytotoxic effects of serum exposure. This was why we included cell counts as a covariate in our analyses (see “model 1”).

"Weaknesses and Considerations: A key limitation lies in the heterogeneity of the ME/CFS patient population, which may obscure subtle serum-induced effects. The wide variability in OCR and ECAR values across individual samples further complicates interpretation. Although the study increased the sample size compared to Fluge et al. (2016), the low representation of severely affected patients may limit the generalizability of the findings, which appear restricted to mild/moderate cases. While stratification by severity did not yield significant differences, it would be valuable to know whether the authors conducted post hoc stratified analyses based on other clinical or biological traits or employed dimensionality reduction techniques such as PCA to explore latent patterns."

We agree that ME symptoms and severity are heterogeneous, a fact that always presents a challenge when studying ME. Specifically, the low representation of severely affected patients may have limited our statistical power to replicate the findings from Fluge et al. (2016). Upon recommendation by the reviewers, we have conducted post hoc stratified analyses of OCR with different disease features (disease duration, trigger, and illness course). We found no evidence for subgroup-specific serum effects based on these variables (see lines 448-453, highlighted in yellow, and new Supplementary Figure 3). We have chosen not to employ dimensionality reduction techniques in this instance, because the sample size for individual clusters would be too small to be informative.

"Data Presentation: In Figures 4D and 4G, the greater dispersion of values in batch 2 for both study groups is noticeable. A brief commentary on potential causes (such as batch effects, sample handling, or donor variability) would help readers better understand this discrepancy and its implications for data interpretation."

We have added a brief comment, as suggested (see lines: 441-443, highlighted in yellow)

"In Figure 4, it appears that 1–2 data points in the healthy control (HC) group may be outliers. It would be helpful if the authors addressed this, either by discussing their impact or clarifying whether any statistical treatment was applied."

All exclusions were decided prior to unblinding. Outliers were excluded if measurements were deemed unreliable due to improbable cell counts (cell counts below 8000 and above 35000), or measurements taken at maximal respiratory capacity were above Q3 + 1.5 x IQR or below Q1 – 1.5 x IQR (see Methods, lines 213-216). However, the scatterplots and violin plots in Figure 4 clearly indicate that the spread of the data is very similar between both groups.

"Tables 1 through 3 could be consolidated into a single comprehensive table to improve readability and reduce redundancy."

Agreed. We have collapsed tables 1-3 into a single table (see “Table 1”), as suggested.

"Figures should include exact p-values, in addition to the current "n.s." (not significant) labels. Consistency in formatting across all figure labels would enhance clarity. There is a typographical error on line 203."

Thank you. We have added the p-values to all figures as suggested, and corrected the typographical error.

Reviewer #2:

"The manuscript presents a well-designed, preregistered replication of the study by Fluge et al. (2016), evaluating the effects of ME/CFS patient serum on mitochondrial function in cultured myoblasts. The study is technically solid, uses a sufficiently powered sample, and applies rigorous statistical methods with proper control of plate effects and randomization. The authors should be commended for their transparency, detailed reporting, and for publishing negative results—an important and often underrepresented aspect of the scientific process. These features make the paper a valuable contribution to the ME/CFS field, which greatly benefits from efforts to test reproducibility under controlled conditions.

However, while the experimental rigor is strong, the narrative emphasis of the paper leans heavily toward discrediting the results of Fluge et al., rather than leveraging this replication outcome to address broader questions about biological variability and cohort standardization in ME/CFS research. A more balanced framing would increase the paper’s constructive impact and relevance beyond a single prior study.

To be specific, the study convincingly shows that no serum-driven mitochondrial phenotype was observed under the specific experimental and cohort conditions tested here. Yet, direct comparability with Fluge et al. remains limited due to several biological and clinical differences between cohorts:

(1) Sex distribution: This study includes only women, while Fluge et al. analyzed both sexes and observed stronger effects in females.

(2) Disease severity and duration: The original cohort contained mostly moderate-to-severe, long-standing ME/CFS cases, while the present cohort is milder, and disease duration is not clearly reported.

(3) Age, medication, fasting state, and comorbidities: These variables are not fully described and may affect circulating metabolites and mitochondrial behavior.

These distinctions suggest that both studies may not be addressing precisely the same biological question, even if the technical protocol is similar. Therefore, it would be more accurate to interpret the current results as indicating no detectable effect within this specific cohort, rather than as a categorical refutation of previous findings."

We agree that biological or clinical differences may exist between the cohorts in the two studies, particularly pertaining to disease severity. Nevertheless, we expect a biological signal with a large effect size (Cohen’s D=1.32), as found in the original study, to be observable upon replication in a study with our statistical power, if it were a generalisable feature of ME. Furthermore, to address the specific points raised:

(1) We intentionally selected only female participants to increase our power to detect any differences. While Fluge et al. (2016) stratified their cohort for performing metabolomic analysis, they did not perform a sex-stratified analysis for the sub-cohort used in the Seahorse experiments and did not report stronger effects of ME serum on OCR in females.

(2) All of our participants were screened using the DecodeME questionnaire and had a clinical diagnosis of ME, and met CCC and/or IoM diagnostic criteria. All of our cases had a disease duration of at least 1-3 years and most of our cases had a disease duration of more than 10 years (see new Supplementary Figure 3). Our study provides greater detail on the cohort characteristics beyond that provided for the subcohort used to perform the Seahorse experiments in Fluge et al.

(3) We explored the relationship between OCR and age in Figure 5C and found no statistically significant correlation (R squared value = 0.0). Information about comorbidities can be found in Supplementary Figure 1B. While we did not include information about fasting status or medications, neither did Fluge et al. (2016) for the sub-cohort on which they performed the experiments that we sought to replicate.

Furthermore, the cohort in Fluge et al. (2016) is subject to potential batch effects: ME cases were collected and stored in 3 batches, whereas healthy controls were collected in 2 further batches, some of which were several years apart:

“62 healthy controls were recruited from blood donors at Haukeland University Hospital, with blood samples taken in 2012. 40 healthy controls were recruited from the staff at the Department of Oncology, Haukeland University Hospital, with blood samples taken in 2015. […]”

Meanwhile, the ME cases were recruited from three separate clinical trials:

“The majority of samples from ME/CFS patients were harvested in late 2014 and 2015 (181 samples from the “RituxME” and “CycloME” trials). The remaining 19 samples were collected in 2010 (in the KTS-2-2010 trial).”

As far as we can see, Fluge et al. (2016) did not account for these batch effects in the Seahorse experiments described in the paper and did not specify which batches the cases and controls originated from. Furthermore, all 102 healthy controls were non-fasting, whereas 47 of the ME cases fasted overnight prior to biobank sampling, while 153 did not. They did not specify whether the 12 ME samples used in their experiments were taken from fasting or non-fasting individuals.

In our study both controls and cases were sampled in each batch and the two sampling batches were acquired only 3 months apart, so that the effect of different storage times and other batch effects were minimised.

"I encourage the authors to revise the Discussion to highlight this key point and to extract a broader methodological message for the field: the necessity of standardized and well-documented cohort characterization (sex, age, disease duration, severity, metabolic state, and preanalytical conditions). Such harmonization would greatly enhance reproducibility and interpretability across future ME/CFS studies."

We agree that there is a need for standardised and well-documented cohort characterisation across the field, and have added this point to the Discussion section (lines 539-541, highlighted in yellow).

"In summary, this manuscript has clear value due to its methodological strength, transparency, and negative results. Reframing the discussion from a refutation to a constructive call for methodological standardization would strengthen the scientific and conceptual contribution of this work."

Reviewer #3: Introduction

"You have introduced the term “people with56 ME (pwME).” Please not that The acronym “pwME” is not universally recognized and may confuse readers unfamiliar with the convention. It is not universal to describe people suffering from other conditions (heart failure etc.) as people with XX. Using such approach might further distant the general public view of the field of ME/CFS from rest of biologically-based severe conditions."

While we share your concern about stigmatisation of ME, and we certainly do not wish to distance ME from biologically-based severe conditions, we chose to use the term “people with ME” (pwME) only after it was recommended by individuals with lived experience of ME. We further involved a patient and public involvement panel in this project who reviewed the manuscript and who did not flag this terminology as a concern.

"Methods

“Sera from pwME and HC were collected between 27/11/2023 - 23/02/2024 across two rounds of sampling over two weeks in November–December 2023 (“batch 1”), and three weeks in February 2024 (“batch 2”) (Table 1).”

Do You think that it might affect results obtained? Please describe it as a potential limitation."

We included sensitivity analyses exploring batch effects in Figure 5D and Figure 5G. We note that in Fluge et al. (2016) batch effects were likely to be more substantial than ours. This is because their patient and control samples came from different sources, some which were collected over a number of years. They do not specify which batches their samples were obtained from, whereas the ME cases and controls we recruited were sampled in both of our batches. We further performed a sensitivity analysis that included sampling batch as a covariate and demonstrated no statistically significant effect of batch on OCR (lines 432-448).

"“Due to the female preponderance of ME (3) and to reduce heterogeneity, all study participants were female. “ Please describe it as a potential limiting factor of the study"

Cohort heterogeneity is a limitation of many studies of ME, including ours. We see the restriction to females as a strength of our study, rather than a limitation, because it minimises one source of potential heterogeneity. Including both sexes would have reduced our statistical power to detect differences, particularly since there is no existing evidence of male-specific effects of ME serum on mitochondrial function. By focussing only on female ME cases, we increased our statistical power.

"“People with ME met the Canadian Consensus Criteria (CCC) and/or the Institute of Medicine (IoM) diagnostic criteria and reported a diagnosis of ME by a healthcare professional. Healthy controls did not meet the CCC or IoM criteria according to their screening survey responses and did not report any of the 21 active comorbidities screened for by the DecodeME screening questionnaire (13).” How many met IOM and how many met CCC? How ME patients and HCs were recruited? Please describe, using flowchart might help"

We have now added this information to the methods (lines 120-121, text highlighted in yellow) and a flowchart to Supplementary Figure 1A describing the screening criteria.

"Discussion:

“Consequently, our study’s results do not support the hypothesis that ME sera impact on healthy myoblast mitochondrial phenotypes differently from healthy control sera.” Can You cite previous studies on blood cell‐based diagnostic test in CFS, that would effectively delineate patients vs controls? What variables were taken into account in those studies?"

We cite in the Introduction several studies that found differences between cells exposed to ME or control serum or plasma (lines 69-80). We focussed on citing and explaining these studies as they were the closest methodologically (predominantly serum swap) and closest in terms of area of biological investigation (predominantly mitochondrial biology). Three additional studies have claimed evidence for blood cell based diagnostic tests in ME/CFS (Esfandyarpour et al. 2019, Xu et al. 2023, Hunter et al. 2025). The fundamental limitation that these papers share is that they are based on modest sample sizes and have not been

---

## [Editor Report · Decision Letter 2]

6 Jan 2026

Indistinguishable mitochondrial phenotypes after exposure of healthy myoblasts to myalgic encephalomyelitis/chronic fatigue syndrome or control serum

PONE-D-25-33599R2

Dear Dr. Ryback,

We’re pleased to inform you that your manuscript has been judged scientifically suitable for publication and will be formally accepted for publication once it meets all outstanding technical requirements.

Kind regards,

Sadiq Umar

Academic Editor

PLOS One

---

## [Editor Report · Acceptance letter]

PONE-D-25-33599R2

PLOS One

Dear Dr. Ryback,

I'm pleased to inform you that your manuscript has been deemed suitable for publication in PLOS One. Congratulations! Your manuscript is now being handed over to our production team.

Kind regards,

on behalf of

Dr. Sadiq Umar

Academic Editor

PLOS One